# An Enhanced Design of Sparse Autoencoder for Latent Features Extraction Based on Trigonometric Simplexes for Network Intrusion Detection Systems

**Hassan Musafer** [1], **Abdelshakour Abuzneid** [1,*], **Miad Faezipour** [1,2,*] **and Ausif Mahmood** [1]

1   Department of Computer Science & Engineering, University of Bridgeport, Bridgeport, CT 06604, USA; hmusafer@my.bridgeport.edu (H.M.); mahmood@bridgeport.edu (A.M.)

2   Department of Biomedical Engineering, University of Bridgeport, Bridgeport, CT 06604, USA

*   Correspondence: abuzneid@bridgeport.edu (A.A.); mfaezipo@bridgeport.edu (M.F.);
    Tel.: +1-203-576-4113 (A.A.); +1-203-576-4702 (M.F.)

**Abstract:** Despite the successful contributions in the field of network intrusion detection using machine learning algorithms and deep networks to learn the boundaries between normal traffic and network attacks, it is still challenging to detect various attacks with high performance. In this paper, we propose a novel mathematical model for further development of robust, reliable, and efficient software for practical intrusion detection applications. In this present work, we are concerned with optimal hyperparameters tuned for high performance sparse autoencoders for optimizing features and classifying normal and abnormal traffic patterns. The proposed framework allows the parameters of the back-propagation learning algorithm to be tuned with respect to the performance and architecture of the sparse autoencoder through a sequence of trigonometric simplex designs. These hyperparameters include the number of nodes in the hidden layer, learning rate of the hidden layer, and learning rate of the output layer. It is expected to achieve better results in extracting features and adapting to various levels of learning hierarchy as different layers of the autoencoder are characterized by different learning rates in the proposed framework. The idea is viewed such that every learning rate of a hidden layer is a dimension in a multidimensional space. Hence, a vector of the adaptive learning rates is implemented for the multiple layers of the network to accelerate the processing time that is required for the network to learn the mapping towards a combination of enhanced features and the optimal synaptic weights in the multiple layers for a given problem. The suggested framework is tested on CICIDS2017, a reliable intrusion detection dataset that covers all the common, updated intrusions and cyber-attacks. Experimental results demonstrate that the proposed architecture for intrusion detection yields superior performance compared to recently published algorithms in terms of classification accuracy and F-measure results.

**Keywords:** intrusion detection system; tuning hyperparameters; Hassan–Nelder–Mead; sparse autoencoder; adaptive learning rates; extracting features; over-fitting problem

## 1. Introduction

### 1.1. Background

As a result of the increasing attacks on Internet-connected devices in recent years, the study of Intrusion Detection Systems (IDS) has attracted strong interests from a wide range of different research communities, including information systems, security-software companies, and computer science fields [1]. An intrusion is defined as a set of actions that violates computer security policies,

such as confidentiality, integrity, and availability [2]. According to the annual report of the Asia Pacific Computer Emergency Response Team (CERT) published in 2018, newly emerging cyber-attacks and threats are evolving with modern technological advances such as artificial intelligence, deep learning, and new trends such as the spreading of Internet of Things (IoT) devices [3]. The main challenge is that attackers are highly skilled programmers and always keep novelty in their tools and techniques with the intent to exploit vulnerabilities in computer systems. Therefore, IDS has become an essential part of network security to monitor and respond to potential intrusions in any computing environment [4].

One of the popular defensive techniques for appropriate intrusion detection and prevention systems is based on machine learning (ML) and artificial neural network (ANN) approaches. These ML- and ANN-based anti-threat systems generate a proactive response to stop attacks before they result in major security incidents [2,5,6]. The advantage of using an ANN is to learn complex non-linear patterns in the input data. When IDS is implemented with ANNs and/or other machine learning algorithms, it provides a computer system with the great advantage of being capable of detecting intrusions and maintaining its security policies effectively [7]. Although IDS using ANN and ML methods are developed to provide optimum response in detecting intrusions, it is still difficult and challenging to detect all kinds of attacks in an efficient and high performance manner [4,7].

### 1.2. Key Contributions

According to the literature [4,8], detection of network attacks is a classification problem because the goal is to clarify whether the packet is normal traffic or a network attack. Therefore, the idea is to capture underlying statistical features of the network traffic data and use them to detect the patterns of malicious attack and normal traffic. In this present work, a novel architecture of IDS based on advanced Sparse Autoencoder (SAE) and Random Forest (RF) is proposed to classify the patterns of the normal packets from those of the network attacks. The proposed architecture trains high dimensional network packets data to produce more mature features that help to achieve a better classification result with supervised learning. Additionally, Hassan–Nelde–Mead (HNM) algorithm [9] is employed to tune the hyperparameters of the SAE with respect to the number of nodes in the hidden layer, learning rate of the hidden layer, and learning rate of the output layer.

### 1.3. Paper Organization

The remainder of this paper is organized as follows. Section 2 briefly glances at prior related work that were developed based on machine and deep learning techniques. Section 3 presents the theory and mathematical model of the proposed architecture. In Section 4, the results of the proposed idea applied on a well-known intrusion dataset is provided. A comparison with other related works is also provided in this section. A discussion is presented in Section 5. Finally, concluding remarks appear in Section 6.

## 2. Related Work

This section briefly glances at intrusion detection algorithms related to the use of the CICIDS2017 dataset [10], which are developed mainly based on machine and deep learning techniques. In the literature, only a few IDS algorithms used sparse autoencoders to extract features based on latent representation concepts. The major challenge comes from the fact that high-level features produced by the traditional SAEs are designed to activate only a few number of the nodes in the hidden layers towards specific attributes of the input instances. This approach of extracting features fails to reflect the relationships of data instances by directly imposing a sparsity constraint in the hidden layers.

In [6,11], the traditional SAE and Support Vector Machine (SVM) have been used as feature extraction techniques while the Random Forest (RF) classifier was applied to detect malicious attacks. The RF is an ensemble learning algorithm that combines bootstrap aggregation with random features selection to create a set of decision trees, which result in a powerful prediction model with controlled variance [12]. In [13], the multilayer perceptron network and payload classifying algorithm (MLP-PC)

was used to distinguish between network intrusions and benign traffic. The MLP network is a deep neural network that consists of five layers and utilizes Adam optimizer. The input layer is composed of 27 nodes, followed by three fully connected hidden layers. Each hidden layer is designed with 64 nodes, dropout probability 0.5, and rectified linear activation function. The output layer is a single node with a sigmoid activation function. The payload classifier (PC) is a deep convolutional neural network (CNN) that consists of a character embedding layer, followed by four convolutional and pooling layers and two standard layers embedded with sigmoid function for classification. In [14], the Fisher Score algorithm (FS) was utilized for feature selection and the SVM, K-Nearest Neighbor (KNN), and Decision Tree (DT) algorithms were applied for intrusion detection, classifying two classes: DDoS or benign. The FS is a supervised feature selection algorithm that selects each feature independently according to a score measured by Fisher criterion [15]. In [16], a distributed model based on Spark was proposed using a collection of a deep belief network (DBN) and multi-layer ensemble support vector machines (MLE-SVMs). The DBN is a greedy layer-wise unsupervised learning model designed with a fine-tuning strategy to learn the relationships among low-level attributes and to represent a good set of hierarchical features. In [17], a deep learning based feature extraction technique and support vector machine (DL-SVM) were used to implement an effective and flexible IDS network. The authors of [18] presented the utilization of RF to keep the most effective features through a recursive feature elimination and deep multilayer perceptron (DMLP) structure to detect intrusion attacks.

## 3. Proposed Methodology

In this section, we present the proposed IDS architecture based on an enhanced SAE and RF algorithm, as depicted in Figure 1. The proposed IDS includes various modules for preprocessing huge amount of network packets, tuning the hyperparameters of SAE, and producing more mature and discriminating features. Typically, the preprocessing module identifies the minimum (Min) and maximum (Max) values of the basic features and normalizes them between 0 and 1. Moreover, features that have one value for different classes are eliminated. The hyperparameters tuning module selects 5% of the network packets, which is used later as features for SAE's offline training to adjust the architecture of the SAE based on the HNM algorithm. The optimizing features module produces fewer features but more mature features and results in improved malicious attacks detection compared to traditional network features. The main modules of the proposed IDS are described in more detail hereafter.

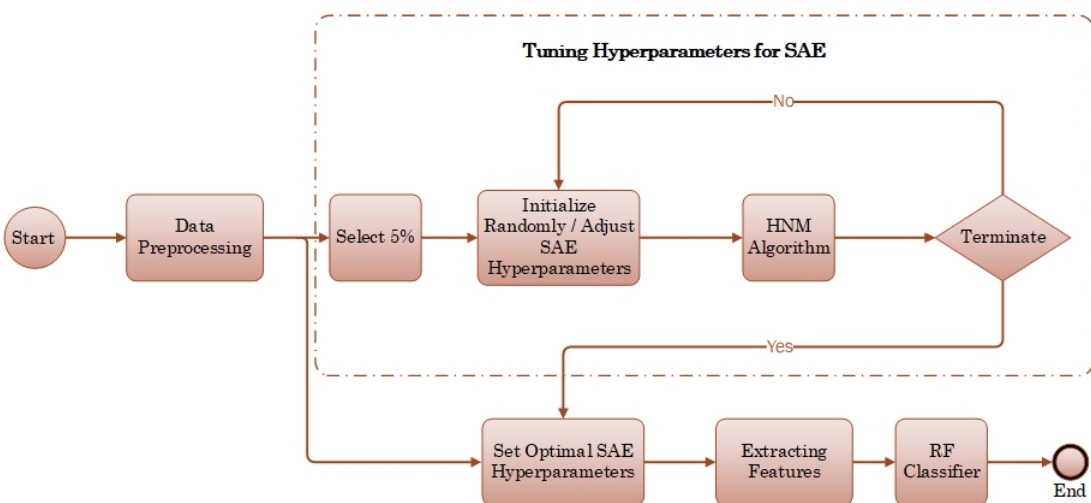

**Figure 1.** Architecture of the proposed IDS based on enhanced SAE and RF.

### 3.1. Data Preprocessing

The data prepossessing module breaks down the Internet Protocol (IP) and port number for sender and receiver, respectively, into four features instead of two; the CICIDS2017 dataset uses

IP-port-sender and IP-port-receiver features. The benefit of doing so is that most intrusions follow a particular pattern for information gathering over the TCP/IP network. After that, the IP-sender and IP-receiver addresses are mapped to an integer representation. Finally, feature scaling is performed to ensure that all the data is in the same range between 0 and 1. Feature scaling is a unity-based normalization method and can be obtained by the following equation [19].

$$x_i = \frac{x_i - x_{min}}{x_{max} - x_{min}} \tag{1}$$

where $x_{min}$ and $x_{max}$ are the minimum and the maximum values for a particular feature $x_i$.

### 3.2. Hassan–Nelde–Mead Algorithm

In this work, we utilize the HNM algorithm [9] to tune the hyperparameters of SAE in order to mitigate the over-fitting problem raised in the hidden layer and to set optimal learning rates for different layers of the back-propagation learning algorithm. The HNM algorithm generates a sequence of trigonometric simplexes designed to extract different features of non-isometric reflections. Unlike the traditional hyperplanes simplex of the Nelder–Mead (NM) algorithm [20,21], the HNM simplex allows the components of the reflected vertex to fragment into multiple triangular simplexes and performs different operations of the algorithm. Thus, the resulting sequence of triangular simplexes not only extracts different non-isometric reflections, but also performs rotation through angles specified by the collection of features of the reflected vertex elements in the hyperplane of the remaining vertices. Therefore, the HNM algorithm is shown to be effective for unconstrained optimization problems. The detailed steps for one axial component of the HNM algorithm is as follows:

Step 1. Initialize Triangular Simplex $(A, B, and\ C)$ and Threshold $(Th)$, as shown in Figure 2:

A tetrahedron simplex is a geometrical object that has three vertices. Each vertex has $n$ components, where $n$ is the dimension of the mathematical problem. Since the HNM algorithm is employed to optimize three hyperparameters of SAE, $n$ in our case equals to 3. Next, we sort the simplex vertices in descending order according to an error function $(E_F)$ that is defined later in the process to obtain four points associated with the lowest, second lowest, second highest, and highest $E_F$ values, such that $A < B < C < Th$. Note, each of $(A, B, C, and\ Th)$ have three axial components (dimensions). The HNM algorithm optimizes a single component in each iteration, while pursuing to explore the curvatures of the $E_F$ through six basic operations.

Step 2. Reflection $D$:

The HNM performs reflection along the line segment that is connecting the worst vertex $C$ and the center of gravity, which is $H$ to evaluate $E_F(D)$. The vector formula for $D$ is given below.

$$D = A + B - C \tag{2}$$

Step 3. Expansion $E$:

If $(E_F(D) < E_F(Th))$, then the HNM executes expansion because it found a descent in that direction (see Figure 2). $E$ is found by the following equation.

$$E = \frac{3A + 3B - 4C}{2} \tag{3}$$

a. If $(E_F(E) < E_F(D))$, then we replace the threshold point $Th$ with $E$, and go to Step 6.
b. Otherwise $Th$ is replaced by $D$, and the algorithm goes to Step 4.

Step 4. Contraction $F$ or $G$:

If $(E_F(D) > E_F(Th))$, then another point must be tested, which is $F$. If $(E_F(F) < E_F(Th))$, then $F$ is kept and replaced with $Th$. If the condition of $F$ is not met, then perhaps a better point is found somewhere between $C$ and the centroid $H$. The point $G$ is computed to see

whether this point has a smaller function value than *Th* or not. The vector formulas for *F* and *G* are as follows.

$$F = \frac{3A + 3B - 2C}{4} \qquad (4)$$

$$G = \frac{A + B + 2C}{4} \qquad (5)$$

a.  If either *F* or *G* has smaller $E_F$ values than *Th*, then *Th* is updated and the algorithm goes to Step 6.

b.  Otherwise, the algorithm moves to Step 5.

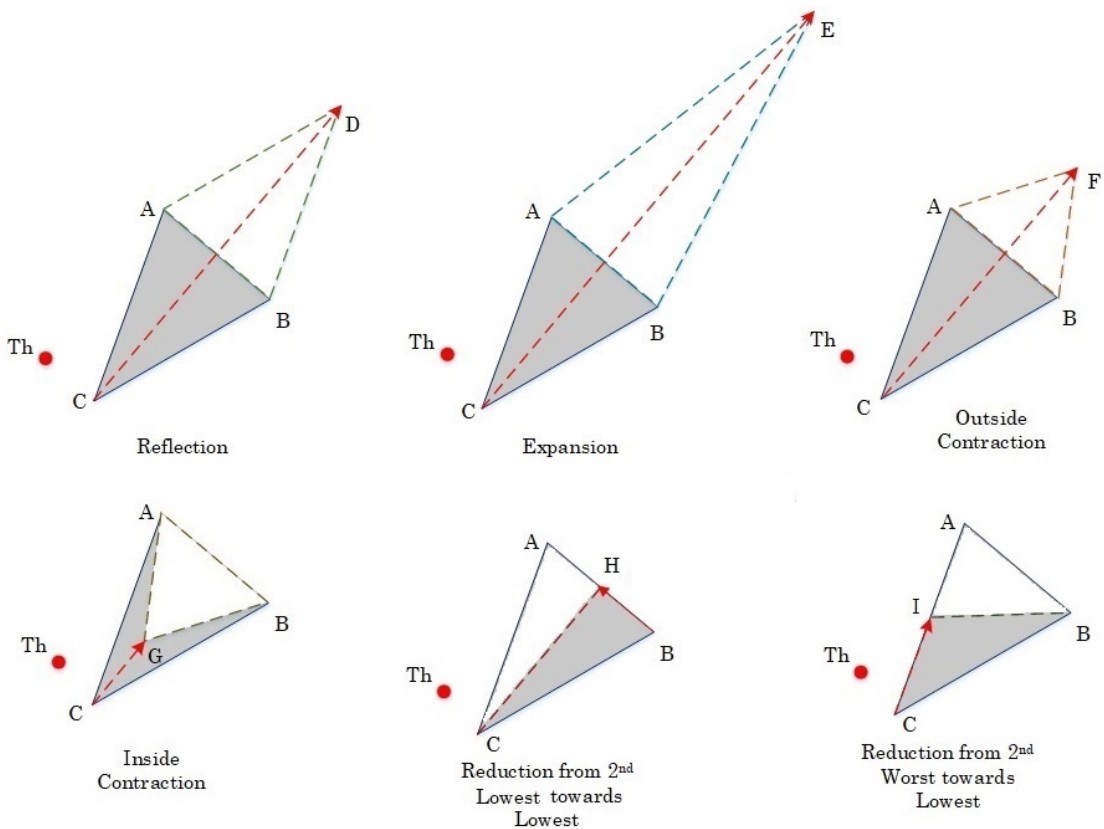

**Figure 2.** The basic operations of the HNM algorithm.

Step 5.  Reduction *H* or *I*:

The HNM algorithm performs two types of shrinkage operations. It shrinks the simplex either at the vertex that has the second lowest $E_F$ value to evaluate *H* or at the second highest vertex to evaluate *I*. The HNM verifies the value of $E_F(H)$. If the condition of point *H* is not satisfied, then HNM shrinks the simplex along the line segment $\overline{AC}$ and evaluates $E_F(I)$. The HNM goes to Step 6. The new vertices are given by:

$$H = \frac{A + B}{2} \qquad (6)$$

$$I = \frac{A + C}{2} \qquad (7)$$

Step 6.  Termination Test:

The termination tests are problem-based and user-defined. In this work, the stopping criterion is primarily characterized by the designed error function of the SAE. It is

encountered in examining the deviation of the error function from the true minimum by $10^{-4}$, as indicated by the inequality below.

$$Stopping\ criterion = \sqrt{\sum_{i=1}^{N}(E_F(x_i, y_i, z_i) - \overline{E_F(x, y, z)})^2} < 10^{-4} \qquad (8)$$

The termination criterion is evaluated for a predefined number of iterations $(N)$. $(x, y, and\ z)$ correspond to the number of nodes in the hidden layer, learning rate of the hidden layer, and learning rate of the output layer, respectively.

If the condition of the termination test is satisfied, the HNM algorithm stops and returns the best architecture of SAE and the learning rates for the different layers of the back-propagation algorithm. Otherwise, the algorithm sorts the simplex vertices and the *Th* and goes to Step 2.

### 3.3. Proposed Sparse Autoencoder

Sparse autoencoder is an unsupervised learning algorithm whose training procedure involves a sparsity penalty, allowing only a few nodes of the hidden layers to activate when feeding a single sample into the network. The intuition behind this idea is that the algorithm is forced to sensitize a small number of individual nodes of the hidden layers towards specific features of the input sample [22,23]. This form of regularization is accomplished by calculating the average activation nodes produced by the hidden layers over a collection of input samples. To satisfy the sparsity constraint, the mean computed over the training samples must be near 0 [22,24]. The main problem, however, is that autoencoders often do not explicitly impose regularization on the weights of the network; instead, they regularize activations. As a result, poor performances are encountered with the early designs of sparse autoencoders such that sparsity makes it difficult for an autoencoder to approximate zero (or near zero) error loss function [24,25].

In contrast to traditional autoencoders, this work proposes an alternative mathematical model for sparse autoencoders, which provides a new platform for developing a compressed feature extraction based on imposing sparsity regularization on the weights, not the activations. One solution to penalize weights within a network would be to impose regularization by the sparsity constraint in the output layer. As a result, the sparse autoencoder is encouraged to find a connection between the sparsity penalty and the learning to extract the latent features by selectively activating the number of variables (weights) of the network. The template of the proposed SAE is illustrated in Figure 3.

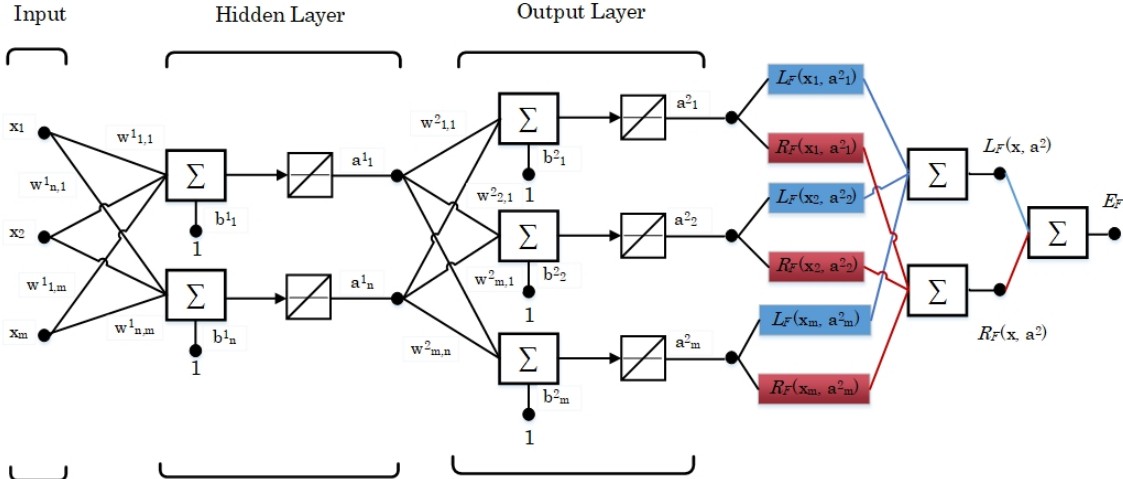

**Figure 3.** Example of sparse autoencoder approximation network.

As discussed above, sparse autoencoder is an unsupervised learning algorithm and relies on conveying the outputs of one layer to become the inputs of the following layer. For *m* input attributes and *n* hidden layer nodes, the equations that describe this operation are as follows.

$$
a^1 = \begin{bmatrix} a_1^1 \\ \vdots \\ a_n^1 \end{bmatrix} = \begin{bmatrix} w_{1,1}^1 & w_{1,2}^1 & \cdots & w_{1,m}^1 \\ \vdots & \vdots & \ddots & \vdots \\ w_{n,1}^1 & w_{n,2}^1 & \cdots & w_{n,m}^1 \end{bmatrix} \begin{bmatrix} x_1 \\ x_2 \\ \vdots \\ x_m \end{bmatrix} + \begin{bmatrix} b_1^1 \\ \vdots \\ b_n^1 \end{bmatrix} \tag{9}
$$

where $w^1 \in R^{n,m}$ is the weight matrix for the hidden layer and $b^1 \in R^{n,1}$ is the bias matrix associated with the hidden layer. As can be seen in Figure 3, the multilayer design of the SAE network has linear activation functions. Thus, the inputs of the output layer are purely represented by the vector $a^1 \in R^{n,1}$.

$$
a^2 = \begin{bmatrix} a_1^2 \\ a_2^2 \\ \vdots \\ a_m^2 \end{bmatrix} = \begin{bmatrix} w_{1,1}^2 & \cdots & w_{1,n}^2 \\ \vdots & \ddots & \vdots \\ w_{m,1}^2 & \cdots & w_{m,n}^2 \end{bmatrix} \begin{bmatrix} a_1^1 \\ \vdots \\ a_n^1 \end{bmatrix} + \begin{bmatrix} b_1^2 \\ \vdots \\ b_m^2 \end{bmatrix} \tag{10}
$$

where $w^2 \in R^{m,n}$ and $b^2 \in R^{m,1}$ are the weight and bias matrices of the output layer. The outputs of the neurons in the last layer are considered as the SAE outputs, which are denoted by the vector $a^2 \in R^{m,1}$.

As shown in Figure 3, the proposed Error Function $(E_F)$ is composed of two different parts. The first term is the Mean Squared Error or Loss Function $(L_F)$ that measures the average squared difference between the estimated (output) and the actual (input) values. The second term is the proposed Regularization Function $(R_F)$, which employs the sparsity constraint, mainly for penalizing the weight matrices of the hidden and output layers. These terms are calculated as follows:

$$
L_F = \frac{1}{m} \sum_{i=1}^{m} (x_i - a_i^2)^2 \tag{11}
$$

$$
R_F = \frac{1}{m} \sum_{i=1}^{m} \left( (x_i + 10) log \frac{x_i + 10}{a_i^2 + 10} + (10 - x_i) log \frac{10 - x_i}{10 - a_i^2} \right) \tag{12}
$$

$$
E_F = L_F + R_F \tag{13}
$$

After propagating the input samples forward through the SAE network and obtaining the output vector $(a^2)$, the next step is to evaluate the $E_F$ from Equation (13). Since $E_F$ is not an explicit function of the weights and bias in the SAE network, we need to specify a sensitivity measure that sensitizes the changes in $E_F$ and propagates these changes backward through the network from the last layer to the first layer, in a process called the back-propagation learning algorithm.

To derive the recurrence relationship for the sensitivities, we use the Stochastic Gradient Descent algorithm (SGD) [26]. For the output layer, the SGD for updating the weight and bias matrices can be expressed as follows.

$$
w^2 = w^2 - \alpha^2 \frac{\partial E_F}{\partial w^2} \tag{14}
$$

$$
b^2 = b^2 - \alpha^2 \frac{\partial E_F}{\partial b^2} \tag{15}
$$

where $\alpha^2$ is the learning rate associated with output layer.

The only complication is that the $E_F$ for a multilayer SAE design is an indirect function of the weights and bias. Thus, the chain rule theory is required to calculate the partial derivatives of $E_F$ with

respect to a third variable such as $w$ or $b$ in the hidden and output layers. By using the chain rule application, the derivatives of Equations (14) and (15) can be simplified to the following:

$$w^2 = w^2 - \alpha^2 \frac{\partial E_F}{\partial a^2} \times \frac{\partial a^2}{\partial w^2} \tag{16}$$

$$b^2 = b^2 - \alpha^2 \frac{\partial E_F}{\partial a^2} \times \frac{\partial a^2}{\partial b^2} \tag{17}$$

We denote the sensitivity at the output layer as $s^2$, which can be defined as:

$$s^2 = \frac{\partial E_F}{\partial a^2} \tag{18}$$

Then, Equations (16) and (17) become:

$$w^2 = w^2 - \alpha^2 \, s^2 \, (a^1)^T \tag{19}$$

$$b^2 = b^2 - \alpha^2 \, s^2 \tag{20}$$

where

$$s^2 = \begin{bmatrix} s_1^2 \\ s_2^2 \\ \vdots \\ s_m^2 \end{bmatrix} = \begin{bmatrix} \frac{-(x_1+10)}{log(10)\,(a_1^2+10)} + \frac{(10-x_1)}{log(10)\,(10-a_1^2)} - \left(x_1 - a_1^2\right) \\ \frac{-(x_2+10)}{log(10)\,(a_2^2+10)} + \frac{(10-x_2)}{log(10)\,(10-a_2^2)} - \left(x_2 - a_2^2\right) \\ \vdots \\ \frac{-(x_m+10)}{log(10)\,(a_m^2+10)} + \frac{(10-x_m)}{log(10)\,(10-a_m^2)} - \left(x_m - a_m^2\right) \end{bmatrix} \tag{21}$$

Following the same procedure for evaluating $s^2$, we can propagate the sensitivities backward from the output layer to the hidden layer as follows.

$$w^1 = w^1 - \alpha^1 \, s^1 \, (x)^T \tag{22}$$

$$b^1 = b^1 - \alpha^1 \, s^1 \tag{23}$$

where $\alpha^1$ is the learning rate associated with the hidden layer and $s^1$ is represented as follows.

$$s^1 = \begin{bmatrix} s_1^1 \\ \vdots \\ s_n^1 \end{bmatrix} = \begin{bmatrix} s_1^2 w_{1,1}^2 + s_2^2 w_{2,1}^2 + \cdots + s_m^2 w_{m,1}^2 \\ \vdots \\ s_1^2 w_{1,n}^2 + s_2^2 w_{2,n}^2 + \cdots + s_m^2 w_{m,n}^2 \end{bmatrix} \tag{24}$$

## 4. Experimental Results

As the rise in attacks on Internet-connected devices are being increased dramatically, it becomes significantly important to consider a reliable dataset that contains volumes of traffic diversity and covers a variety of attacks. Following this trend, we tested our proposed IDS architecture on the CICIDS2017 dataset that covers almost the all common updated attacks such as DDoS, DoS, SQL Injection, Brute Force, XSS, Botnet, Infiltration, and Port Scan attacks. In addition, this section presents two experimental results in examining the efficiency and reliability of the proposed SAE network and shows comparisons with other relevant works. While mitigating the effect of the over-fitting problem, we used the HNM algorithm to determine the number of nodes in the hidden layer based on the initial values of weights and bias in the network.

As shown in Figure 1, data preprocessing is the first step of preparing the records of the dataset, which includes unity-based normalization and eliminating the attributes that have one value in all instances of the dataset. After preprocessing, the volume of the dataset was reduced to 70 features. Then, at least 5% of the reduced dataset was randomly selected to be used later by the HNM algorithm. The aim of using the HNM algorithm was to tune the hyperparameters of the SAE architecture, optimize the learning rates for the different layers, and set percentages of the sparsity for the different layers. Because the weights and bias values were initialized randomly, tuning the hyperparameters for the IDS design differed from one iteration to another. In this paper, we report two experiments to observe how the hyperparameters are tuned based on random initialization and the results are summarized in Table 1. All experiments and simulations were carried out using an Intel-Xeon processor with 3.70 GHz and 16 GB RAM, running Windows 10.

**Table 1.** Hyperparameters of the proposed SAE network.

| Experiment | $n$ | $\alpha^1$ | $\alpha^2$ | $S_P^1$ | $S_P^2$ | Epoch | Time (s) |
|---|---|---|---|---|---|---|---|
| First | 12 | 0.01037246 | 0.1032246 | 22% | 32% | 180 | 4091 |
| Second | 5 | 0.00173640 | 0.0524296 | 22% | 34% | 126 | 2394 |

As illustrated in Table 1, different parametric measures are produced corresponding to the first and second experiments. These hyperparameters include: number of nodes in the hidden layer ($n$), learning rate in the hidden layer ($\alpha^1$), learning rate in the output layer ($\alpha^2$), percentage of sparsity measured for the hidden layer ($S_P^1$), percentage of sparsity measured for the output layer ($S_P^2$), number of epochs (Epoch), and time in seconds (Time). Additionally, it can be seen that the values of the learning rates can be made to vary from one layer to another. This gives us a better features extraction strategy, where the different layers can adapt to various levels of the learning hierarchy. This is while the percentages of sparsity, which are computed for the weight matrices, remain almost stable for both of the conducted experiments.

The proposed SAE for IDS applications was implemented in C# language and tested on the CICIDS2017 dataset containing about 2,830,235 instances. After fine-tuning was performed using the HNM algorithm, the Random Forest (RF) classifier was used as the last layer to detect/distinguish fifteen classes, including the different types of attack packets and the normal traffic packets. Table 2 summarizes the test results in terms of F-measure ($F_M$) and accuracy ($Acc$) to cover the experiments entirely. The table also shows comparisons with our previous work and with some of the recently published algorithms. Three criteria are reported to characterize the different algorithms: number of classes ($Cs$), training set ($Tr$), and testing set ($Ts$). Finally, the feature extraction method and number of features are recorded from the corresponding papers in the last column of the table. The * symbol indicates that the performance measure has not been reported.

**Table 2.** Summary of the experimental results and comparison with other techniques.

| Algorithm | Experiment Criteria | Performance Measure | Feature Extraction |
|---|---|---|---|
| 1. MLP-PC [13] | ($Cs = 9$) ($Tr = 50\%$) ($Ts = 50\%$) | ($F_M = 0.948$) ($Acc = *$) | (27) |
| 2. KNN [14] | ($Cs = 2$) ($Tr = 90\%$) ($Ts = 10\%$) | ($F_M = 0.990$) ($Acc = 99.0\%$) | FS (30) |
| 3. MLE-SVMs [16] | ($Cs = 15$) ($Tr = 60\%$) ($Ts = 40\%$) | ($F_M = 0.926$) ($Acc = *$) | DBN (16) |
| 4. DL-SVM [17] | ($Cs = 2$) ($Tr = 67\%$) ($Ts = 33\%$) | ($F_M = 0.990$) ($Acc = 97.8\%$) | (85) |
| 5. DMLP [18] | ($Cs = 2$) ($Tr = 80\%$) ($Ts = 20\%$) | ($F_M = *$) ($Acc = 91.0\%$) | RF (10) |
| 6. RF [6] | ($Cs = 15$) ($Tr = 70\%$) ($Ts = 30\%$) | ($F_M = 0.995$) ($Acc = 99.1\%$) | SAE (59) |
| 7. **SAE-RF** | ($Cs = 15$) ($Tr = 70\%$) ($Ts = 30\%$) | ($F_M = $**0.996**) ($Acc =$**99.63**%) | **Proposed SAE (12)** |
| 8. **SAE-RF** | ($Cs = 15$) ($Tr = 70\%$) ($Ts = 30\%$) | ($F_M = $**0.996**) ($Acc =$**99.56**%) | **Proposed SAE (5)** |

## 5. Discussion

As demonstrated in Table 2, the two conducted experiments achieved results that outperform the existing solutions introduced for the updated and different types of network attacks. Thereby,

the proposed SAE architecture provides better performance to extract a good set of features, which could reveal high levels of representation towards various characteristics of the latest intrusion attacks. This is proven by the test results. The features produced by the enhanced SAE technique had learned latent representation to sensitize the individual synaptic weights in the hidden layer and to generate keys for better classification accuracy and F-measure results. The measurements of true positive rate, false positive rate, precision, recall, total number of epochs required to extract the latent features, and time in seconds (Time) for both experiments are summarized in Table 3. After tuning hyperparameters of the improved SAE, it required 3925 s to discover 12 latent features for the first experiment and 2034 s to discover five latent features for the second experiment based on random initialization. Even though the second experiment took less time to represent the latent features, it failed to provide better performance in terms of the accuracy and false positive rate.

**Table 3.** Details of the results for two experiments.

| Experiment | True Positive Rate | False Positive Rate | Precision | Recall | Epoch | Time (s) |
|---|---|---|---|---|---|---|
| First | 0.996 | 0.009 | 0.996 | 0.996 | 76 | 3925 |
| Second | 0.996 | 0.011 | 0.996 | 0.996 | 27 | 2034 |

## 6. Conclusions

This paper proposes an enhanced design of the SAE architecture for IDS applications. The proposed error function for the SAE is designed to make a trade-off between the latent state representation for more mature features and network regularization by applying the sparsity constraint in the output layer of the proposed SAE network. In addition, the hyperparameters of the SAE are tuned based on the HNM algorithm and were proved to give a better capability of extracting features in comparison with the existing developed algorithms such as MLP-PC, MLE-SVMs, and DMLP. In fact, the proposed SAE can be used for not only network intrusion detection systems, but also other applications pertaining to feature extraction and pattern analysis. We emphasize that the successful contribution of allocating a set of optimal learning rates for different layers of the proposed SAE network has resulted in developing an efficient SAE architecture that can be used to discover latent features extraction. The results from experimental tests show that the different layers of the enhanced SAE could efficiently adapt to various levels of the learning hierarchy. Besides, this provided the SGD algorithm with the ability to dynamically adjust the weights and biases within the network. It is evidenced from the test results that it is possible to accelerate the learning process to reach latent features representation based on a vector of adaptive learning rates applied for the multiple layers of the proposed SAE network. Finally, additional tests demonstrated that the proposed IDS architecture could provide a more compact and effective immunity system for different types of network attacks with a significant detection accuracy of 99.63% and an F-measure of 0.996, on average, when penalizing sparsity constraint directly on the synaptic weights within the network.

**Author Contributions:** Supervision, A.M., A.A. and M.F.; Writing—original draft preparation, H.M.; Writing—review and editing, M.F. and A.A.; Conceptualization, H.M. and A.M.; Methodology, H.M., M.F., and A.A.; Software, H.M.; Validation, M.F. and A.A.; Formal Analysis, H.M., M.F., and A.A.; Investigation, A.A. and M.F.; Resources, A.A., M.F., and A.M.; Data Curation, H.M.; Visualization, H.M., M.F., and A.A.; and Project Administration A.A. and M.F. All authors have read and agreed to the published version of the manuscript.

**Funding:** This research was funded in part by the UB Seed Money Grant 2018 and the UB Partners CT Next Innovation Grant 2018-2020.

**Conflicts of Interest:** The authors declare no conflict of interest.

## Abbreviations

The following abbreviations are used in this manuscript:

| | |
|---|---|
| *Acc* | Accuracy |
| ANN | Artificial Neural Network |
| CNN | Convolutional Neural Network |
| *Cs* | Number of Classes |
| DBN | Deep Belief Network |
| DDoS | Distributed Denial-of-Service |
| DL-SVM | Deep Learning - Support Vector Machine |
| DMLP | Deep Multi-layer Perceptron |
| DoS | Denial-of-Service |
| DT | Decision Tree |
| $E_F$ | Error Function |
| $F_M$ | F-measure |
| HNM | Hassan–Nelde–Mead |
| IDS | Intrusion Detection Systems |
| KNN | K-Nearest Neighbor |
| $L_F$ | Loss Function |
| ML | Machine Learning |
| MLE-SVMs | Multi-layer Ensemble Support Vector Machines |
| MLP-PC | Multi-layer Perceptron-Payload Classifier |
| RF | Random Forest |
| $R_F$ | Regularization Function |
| SAE | Sparse Autoencoder |
| SGD | Stochastic Gradient Descent |
| SQL | Structured Query Language |
| *Tr* | Training set |
| *Ts* | Testing set |
| XSS | Cross-site Scripting |

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
