# Peer review of "An Enhanced Design of Sparse Autoencoder for Latent Features Extraction Based on Trigonometric Simplexes for Network Intrusion Detection Systems"

_electronics, doi:10.3390/electronics9020259_

Round 1

Reviewer 1 Report

This paper proposed a design of sparse autoencoder for network intrusion detection system (IDS). The authors used latent features extraction based on trigonometric simplexes for their proposed design. The proposed IDS and sparse autoencoder are well explained in section 3. In section 4, the authors compared the proposed method with other popular methods such as convolutional neural network (CNN) and deep belief network (DBN). But the authors did not illustrate the detail architectures of this methods, because it is important to show the performances.

Author Response

Summary of Modifications

We received valuable feedback from the respectful reviewers. They significantly helped us to improve the quality of our manuscript. For that, we would like to thank the editor, associate/assistant editor and the reviewers. In summary, the changes applied to the manuscript based on the reviewers’ comments, are as follows:

All the reviewers’ comments and suggestions have been carefully considered and addressed. A few references were added for better clarity. The revised reference list has been re-ordered and re-numbered. Explanations have been added/modified according to the reviewers’ comments and/or the authors’ observations. All the changes applied have been highlighted in the revised manuscript.

Now we explain the changes in detail per item. The items are categorized per reviewer. Our response to each item starts with ‘R:’ and is shown in boldface. 

We are truly grateful to have been given the chance to revise the paper according to the reviewers’ comments. In what follows, we have provided the reviewers’ comments and our responses regarding how we have addressed each of the comments. We have made our best efforts to address all comments. Thanks so much.

Reviewer #1:

Comments to the Author:

Q. This paper proposed a design of sparse autoencoder for network intrusion detection system (IDS). The authors used latent features extraction based on trigonometric simplexes for their proposed design. The proposed IDS and sparse autoencoder are well explained in section 3. In section 4, the authors compared the proposed method with other popular methods such as convolutional neural network (CNN) and deep belief network (DBN). R:

R: Thank you very much. We are glad and grateful that the reviewer found our paper satisfying.

Q. The authors did not illustrate the detail architectures of this methods, because it is important to show the performances.

R: We thank the reviewer for this useful suggestion. We have provided a brief analysis of each of the compared algorithms to emphasize the contributions against the updated intrusions and cyber-attacks (CICIDS2017). References have been re-ordered and additional descriptions are included in the revised manuscript:

·         In [6, 11], the traditional SAE and support vector machine (SVM) have been used as feature extraction techniques while the random forest (RF) classifier was applied to detect malicious attacks. The RF is an ensemble learning algorithm that combines bootstrap aggregation with random features selection to create a set of decision trees which result in a powerful prediction model with controlled variance [12].

·         In [13], the multilayer perceptron network and payload classifying algorithm (MLP-PC) was used to distinguish between network intrusions and benign traffic. The MLP network is a deep neural network that consists of five layers and utilizes Adam optimizer. The input layer is composed of 27 nodes, followed by three fully connected hidden layers. Each hidden layer is designed with 64 nodes, dropout probability 0.5, and rectified linear activation function. The output layer is a single node with a sigmoid activation function. This is while the payload classifier (PC) is a deep convolutional neural network (CNN) that consists of a character embedding layer, followed by four convolutional and pooling layers and two standard layers embedded with sigmoid function for classification.

·         In [14], the Fisher Score algorithm (FS) was utilized for feature selection and the SVM, K-Nearest Neighbor (KNN) and Decision Tree (DT) algorithms were applied for intrusion detection, classifying two classes:  DDoS or benign. The FS is a supervised feature selection algorithm that selects each feature independently according to a score measured by Fisher criterion [15].

·         In [16], a distributed model based on Spark was proposed using a collection of a deep belief network (DBN) and multi-layer ensemble support vector machines (MLE-SVMs). The DBN is a greedy layer-wise unsupervised learning model designed with a fine-tuning strategy to learn the relationships among low-level attributes and to represent a good set of hierarchical features.

Reviewer 2 Report

The authors proposed a network intrusion detection framework to learn the boundaries between normal traffic and network attacks. The proposed approach yields a good performance comparing to existing classification algorithms in terms of classification accuracy and F-Measure.

The authors covered/included all the sections to be published this work as MDPI article. However, the related work section is poorly presented. The authors can include more relevant works and compare them with the proposed approach. Please consider a recent research work on "effective analysis of machine learning classification models...".

Author Response

Summary of Modifications

We received valuable feedback from the respectful reviewers. They significantly helped us to improve the quality of our manuscript. For that, we would like to thank the editor, associate/assistant editor and the reviewers. In summary, the changes applied to the manuscript based on the reviewers’ comments, are as follows:

All the reviewers’ comments and suggestions have been carefully considered and addressed. A few references were added for better clarity. The revised reference list has been re-ordered and re-numbered. Explanations have been added/modified according to the reviewers’ comments and/or the authors’ observations. All the changes applied have been highlighted in the revised manuscript.

Now we explain the changes in detail per item. The items are categorized per reviewer. Our response to each item starts with ‘R:’ and is shown in boldface. 

We are truly grateful to have been given the chance to revise the paper according to the reviewers’ comments. In what follows, we have provided the reviewers’ comments and our responses regarding how we have addressed each of the comments. We have made our best efforts to address all comments. Thanks so much.

Reviewer #2:

Comments to the Author:

Q. The authors proposed a network intrusion detection framework to learn the boundaries between normal traffic and network attacks. The proposed approach yields a good performance comparing to existing classification algorithms in terms of classification accuracy and F-Measure.

R: Thank you very much. We are glad and grateful that the reviewer believes our paper displays a strong contribution.

Q. However, the related work section is poorly presented. The authors can include more relevant works and compare them with the proposed approach. Please consider a recent research work on "effective analysis of machine learning classification models...”

R: We thank the reviewer for this useful suggestion.

·         We have carefully revised the related works section to include the reviewer’s points.

·         Two additional references were added and cited in the revised related work for more detailed explanations on the prior related work regarding their algorithms, structure and performances:

 [12] Sarker IH, Kayes AS, Watters P. Effectiveness analysis of machine learning classification models for predicting personalized context-aware smartphone usage. Journal of Big Data 2019, 6(1), 57.

[15] Gu Q, Li Z, Han J. Generalized fisher score for feature selection. arXiv preprint arXiv: 1202.3725. 2012.
